

**Wintertime New Particle Formation and Its Contribution to Cloud Condensation**
**Nuclei in the Northeastern United States**
Fangqun Yu[1], Gan Luo[1], Arshad Nair[1], James J. Schwab[1], James P. Sherman[2], and Yanda Zhang[1]
[1]Atmospheric Sciences Research Center, State University of New York, Albany, New York 12203, USA
[2]Department of Physics and Astronomy, Appalachian State University, NC 28608, USA
**Abstract**: Atmospheric particles can act as cloud condensation nuclei (CCN) and modify cloud
properties and precipitation and thus indirectly impact the hydrological cycle and climate. New
particle formation (NPF or nucleation), frequently observed at locations around the globe, is an
important source of ultrafine particles and CCN in the atmosphere. In this study, wintertime NPF
over the Northeastern United States (NEUS) is simulated with WRF-Chem coupled with a size-
resolved (sectional) advanced particle microphysics (APM) model. Model simulated variations of
particle number concentrations during a two-month period (November–December 2013) are in
agreement with corresponding measurements taken at Pinnacle State Park (PSP), New York and
Appalachian State University (APP), North Carolina. We show that even during wintertime,
regional nucleation occurs and contributes significantly to ultrafine particle and CCN number
concentrations over the NEUS. Due to low biogenic emissions during this period, wintertime
regional nucleation is solely controlled by inorganic species and the newly developed ternary ion-
mediated nucleation scheme is able to capture the variations of observed particle number
concentrations (ranging from ~ 200 – 20,000 cm$^{-3}$) at both PSP and APP. Total particle and CCN
number concentrations dramatically increase following NPF events and have highest values over
the Ohio Valley region, where elevated [$SO_2$] is sustained by power plants. Secondary particles
dominate particle number abundance over the NEUS and their fraction increases with altitude
from >~85% near surface to >~95% in the upper troposphere. The secondary fraction of CCN also
increases with altitude, from 20–50% in the lower boundary layer to 50–60% in the middle
troposphere to 70–85% in the upper troposphere. This significant contribution of wintertime
nucleation to aerosols, especially those that can act as CCN, is important considering the changing
paradigm of wintertime precipitation over the NEUS.



## 1. Introduction

Particle number concentration is a key parameter important for the health and climate impacts of atmospheric aerosols. High number concentrations of ultrafine particles may lead to adverse health effects (Knibbs et al., 2011; Han et al., 2016). Variations in the number concentration of cloud condensation nuclei (CCN) influence cloud properties and precipitation and thus indirectly affect the hydrological cycle and climate (e. g, Twomey, 1977; Charlson et al., 1992). Aerosol particles appear in the troposphere due to either in-situ new particle formation (NPF, i.e, formation of secondary particles (SP) via nucleation) or direct emissions (i.e., primary particles (PP)). Though NPF has little effect on the total particle mass in the immediate vicinity of the nucleation itself, it is highly relevant to the aerosol health and climate effects as SP can dominate the ultrafine particles and those particles that can act as CCN (Spracklen et al., 2008; Pierce and Adams, 2009; Yu and Luo, 2009). Aerosol number concentrations exhibit significant spatial and temporal variability due to non-linear dependence of NPF rates on atmospheric conditions and concentrations of gaseous precursors, both of which are subject to changes as a result of climate changes and emission regulatory actions.

Laboratory experiments and theoretical studies indicate that sulfuric acid, ammonia, amines, ions, and certain organic compounds can all contribute to NPF (see recent review paper by Lee et al., 2019). However, the actual contribution of various nucleation pathways and key controlling parameters in the real atmosphere remains elusive, especially with regard to the relative importance of inorganic versus organic nucleation (e.g., Yu et al., 2015). Inorganic and organic nucleation precursors have quite different sources and their emission strengths depend on different factors, with important implications to spatial distributions of NPF and CCN and their short-term (diurnal, seasonal) and long-term (pre-industry, present, and future climate and emissions) variations. Both inorganic and organic nucleation schemes are subject to uncertainties and it is important to evaluate their ability to capture particle formation and variations of number concentration in the atmosphere. Yu et al. (2015) showed that both inorganic nucleation and organic mediated nucleation can explain NPF observed in a spring month at several forest sites in North America but organic-mediated nucleation over-predicted NPF in the summer.

The main objective of the present study is to investigate the new particle formation process and its contribution to particle number concentration and CCN in the wintertime in the Northeastern United States (NEUS). Wintertime biogenic emissions are very low in NEUS and





thus the contribution of biogenic organic species to NPF is expected to be negligible, enabling us to unequivocally evaluate the performance of the inorganic nucleation scheme. In addition to delineating the underlying processes controlling particle number concentrations in the atmosphere, an improved understanding of major sources and concentrations of CCN in wintertime is also important for better forecasting wintertime precipitation, such as snow storms, in NEUS (Gaudet et al., 2019).

## 2. Methods

2.1 Model

We employ WRF-Chem (version 3.7.1), a regional multi-scale meteorology model coupled with online chemistry (Grell et al., 2005). The model configurations include Morrison 2-mom microphysics (Morrison et al., 2009), RRTMG longwave and shortwave radiation (Clough et al., 2005), Noah land surface, Grell-3 cumulus (Grell and Freitas, 2014), and YSU PBL scheme (Hong et al., 2006). We use CB05 scheme (Yarwood et al., 2005) for gas-phase chemistry, SORGAM with aqueous reactions (Schell et al., 2001) for secondary organic aerosol chemistry and aqueous phase chemistry, and ISORROPIA II (Fountoukis and Nenes, 2007) for aerosol thermodynamic equilibrium. The initial and boundary conditions for meteorology are generated from the National Centers for Environmental Prediction (NCEP) Final (FNL) with resolution at $1° \times 1°$ and time intervals at six hours. The anthropogenic emissions are based on the Environmental Protection Agency's (EPA) National Emission Inventory (NEI) 2011, and the biogenic emissions are calculated using MEGAN (Guenther et al., 2006). Annual scaling factors for NOx, $SO_2$, $NH_3$, and CO derived from EPA's Air Pollutant Emissions Trends Data from 1990 to 2016 are used here to scale the emissions of corresponding species from the baseline year of 2011 to the simulation year. We also considered seasonal variation of $NH_3$ emission due to agricultural activity in the model.

For particle microphysics, we use a size-resolved (sectional) advanced particle microphysics (APM) model (Yu and Luo, 2009) that was previously integrated into WRF-Chem v3.1.1 (Luo and Yu, 2011). For this study, we have updated APM and integrated it into WRF-Chem v3.7.1. Major changes to APM include: (1) employment of 15 bins to represent black carbon (BC) and another 15 bins to represent primary organic carbon (POC) particles in the size range of 3 nm to 2 μm (instead of two log-normal modes in the previous version); (2) consideration of the successive oxidation aging of secondary organic gases (SOG) and explicit kinetic condensation of low volatile




SOG onto particles following the scheme of Yu (2011); (3) fully coupled APM aerosols with WRF-Chem radiation code and cloud microphysics, with aerosol optical properties and aerosol activation calculated from size-resolved APM aerosols using optical properties lookup tables (Yu et al., 2012) and the activation scheme of Abdul-Razzak and Ghan (2002). Cloud droplet number predicted by APM directly impacts spectral shape parameter and slope parameter for cloud droplets in the Morrison 2-mom microphysics scheme and then impacts cloud droplet effective radius, the auto-conversion of cloud water to rainwater, and ultimately affects the rainwater mass content and raindrop number concentration.

We have carried out WRF-Chem-APM simulations for the period of October 25 – December 31, 2013 at 27 km × 27 km horizontal resolution. The domain covered the main continental United States, extending approximately from latitudes 21° N to 54° N and from longitudes 62° W to 132° W, with 180 grid nodes in the east–west direction and 126 in the north–south direction. The model has 30 vertical layers from the surface to 5 hPa, with finer resolution near the surface (6 layers within ~1 km above surface). The simulations were restarted on November 1, November 16, December 1, and December 16, 2013 with continuous chemistry fields from previous runs. The present analysis focuses on the NEUS during November and December of 2013. Simulated 3-D fields meteorological, chemical, and aerosol variables were output every three hours for each grid box and every 15 minutes at the measurement sites described below.

2.2 Measurement site description

2.2.1. Pinnacle State Park (PSP), Addison, New York (NY)

The PSP site is located in Addison, NY, a village in southwestern NY. Its coordinates are 42.09°N and 77.21°W, and it is about 504 meters (m) above sea level (Schwab et al., 2009). The area surrounding PSP contains a variety of vegetation, including a golf course to the northwest; forestlands consisting of deciduous and coniferous trees; pastures and fields; and a 50-acre pond to the site's south (Schwab et al., 2009). The two nearest population centers to PSP are Addison and Corning. The village of Addison is about 4 km to the northwest of PSP, and it has a population of approximately 1800 people. The city of Corning is about 15 km to the northeast of PSP, and it has a population of approximately 11,000 people. Parameters measured include particle number concentration with a TSI model 3783 CPC, $SO_2$ with a Thermo model 43i, temperature, relative humidity, wind speed and direction, solar radiation, and precipitation with calibrated



meteorological sensors. These data are collected as minute averages. Gaseous $NH_3$ is collected as
part of the AMon network as passive two week samples from the nearby Connecticut Hill site
(NADP, 2018).

2.2.2. Appalachian State University (APP), Boone, North Carolina (NC)

The APP site is located at 1076 m on a hill overlooking the campus of Appalachian State

University (Boone, NC) in the heart of the Southern Appalachian Mountains (36.2° N, 81.7° W)
(Sherman et al., 2015). The APP site is surrounded by forests in all directions and is not located
near any major highways or major industry. The Charlotte metropolitan area (population 2.5
million) is located approximately 160 km SE of APP and the Piedmont Triangle metropolitan area
(population 1.6 million) is located 200–230 km ESE of APP. Aerosol optical and microphysical
properties are measured as part of NOAA Earth System Research Laboratory (ESRL) (Sherman et
al., 2015).

**3. Results**

WRF-Chem-APM simulated wintertime NPF over the NEUS for the two-month period

(November–December 2013) is examined. The nucleation rate is calculated with a recently
developed $H_2SO_4$-$H_2O$-$NH_3$ ternary ion-mediated nucleation (TIMN) scheme (Yu et al., 2018),
which is supported by the detailed CLOUD (Cosmics Leaving OUtdoor Droplets) measurements
(Kirkby et al., 2011; Kurten et al., 2016). According to the TIMN scheme, $H_2SO_4$ and $NH_3$ are key
nucleation precursors and other parameters such as temperature, relative humidity, ionization rate,
and surface area of pre-existing particles also influence nucleation rates. $H_2SO_4$, well recognized
to be critical for NPF in the atmosphere, is the oxidation product of $SO_2$. Figure 1 shows the
modeled horizontal spatial distribution for the lower boundary layer (first three model layers, ~ 0
– 400 m above surface) over NEUS during November–December 2013 of the concentrations of
major aerosol precursors (a) $SO_2$ & (b) $H_2SO_4$, and (c) $NH_3$, (d) nucleation rate (J), (e) number
concentration of condensation nuclei > 10 nm (CN10), and (f) number concentration of CCN at
supersaturation 0.4% (CCN0.4). Typical wintertime modeled concentrations of aerosol precursors
in the lower boundary layer over the NEUS are $[SO_2]$ ~ 0.3 – 2 ppbv, $[H_2SO_4]$ ~ 0.03 – 0.2 pptv,
and $[NH_3]$ ~ 0.1 – 5 ppbv. The modeled spatial distribution of the aerosol precursors is co-located
with their source regions: $SO_2$ distribution is in line with the NEI and indicative of coal-fired power





plants in the region, especially over the Ohio Valley. $NH_3$ hotspots are over emission regions of
agricultural land-use and concentrated animal feeding operations. Calculated monthly mean
nucleation rates in the lower boundary layer range typically from ~ 0.1 to ~ 2 $cm^3s^{-1}$ over the NEUS
domain and spatial distributions are strongly correlated with concentration of aerosol precursors,
with negligible nucleation over the oceanic area off the east coast. The number concentrations of
CN10 and CCN0.4, calculated from the simulated particle number size distributions, are ~ 2000–
7000 $cm^{-3}$ and ~ 100–1000 $cm^{-3}$, respectively. Both CN10 and CCN0.4 have highest values over
the Ohio Valley region.
To develop further confidence in WRF-Chem-APM simulations, diurnal variations of these
aerosol precursors, as well as meteorological factors are compared with available in situ
measurements for this two-month period at the PSP site in Figure 2. The meteorological parameters
compared are temperature (T) at 2 m above surface, relative humidity (RH), wind direction, solar
radiation, and precipitation in Figure 2 (a–c). Overall, WRF-Chem-APM simulates the diurnal
variations of T and RH in good agreement with measurements (Fig. 3a), with Pearson correlation
coefficient ($r$) of 0.93 for hourly T and 0.74 for hourly RH. The model also captures major changes
in wind direction (Fig. 2b), solar radiation (Fig. 2b), and occurrence of precipitation (Fig. 2c). The
model slightly over-predicted RH and T. It should be noted that RH measurements were taken at
2 m above surface while modeled RH is the average of model surface layer (~ 0–100 m). The
differences/deviations during some days can also be associated with model uncertainties and sub-
grid variations within the 27 km × 27 km grid box. In situ measurements of [$SO_2$] and [$NH_3$] from
the PSP site are used to examine their simulated values. Absolute values of [$SO_2$] and their day-
to-day variations (from below 0.1 ppbv to above 1 ppbv) are overall consistent with observations
(Fig. 2c), with $r$ of 0.48 and mean bias error (MBE) of −12%. The daily variation of [$NH_3$] (Fig.
2d) is more dramatic than that of [$SO_2$], with the maximum value reaching ~ 10 ppbv on Day 320
and minimum value approaching zero on many days. In WRF-Chem, [$NH_3$] is calculated with
ISOROPIA II (Fountoukis and Nenes, 2007) and assumes equilibrium between gaseous and
particulate phases. In addition to emission, deposition, and transport, [$NH_3$] is also controlled by
particle compositions and temperature. The best available [$NH_3$] data for the site during this period
is from the Ammonia Monitoring Network (AMoN), which provides 2-week averages. The
average values of modeled (observed) [$NH_3$] during November and December are 0.26 (0.5) and
0.04 (0.2) ppbv, respectively, indicating average model–observation consistency with lower bias



of model simulations. Measurements of [NH₃] at high temporal resolution are apparently needed
to more rigorously evaluate the model performance.
During this wintertime period, biogenic emissions are low, leading to negligible modeled
isoprene and monoterpene (not shown) and [LV-SOG] (Fig. 2d, generally $< 10^6$ cm$^{-3}$). In contrast,
the peak [H₂SO₄] can reach above $10^7$ cm$^{-3}$. As a result of its sole production from photochemistry
and its short lifetime associated with condensation on pre-existing particles, [H₂SO₄] shows strong
diurnal variation. [H₂SO₄] above $\sim 3\times10^6$ cm$^{-3}$ is a necessary condition for substantial nucleation
(with nucleation rate $J > 0.1$ cm$^{-3}$ s$^{-1}$) to occur (Fig. 2e). On Days 319 and 320 (November 15-16),
peak [H₂SO₄] was above $3\times10^7$ cm$^{-3}$ and maximum nucleation rate reached up to 10 cm$^{-3}$ s$^{-1}$. In
addition to [H₂SO₄], which also depends on surface area of pre-existing particles (and hence RH),
[NH₃] and T are other two important parameters controlling the variations of nucleation rates. It
should be noted that ionization rates assumed in the model, while also important for NPF under
the conditions, do not have much temporal and horizontal variations. The variations of $J$ lead to
large changes of CN10, from several hundreds to above tens of thousands per cm$^{-3}$, which is in
good agreement with observations (Fig. 2e) and analyzed in more detail in Figure 3.
Figure 3 presents simulated surface-level (model first layer) particle number size distributions
(PNSD), and CN10, and CCN0.4 during the two-month period for two sites in NEUS where CN10
in situ measurements are available: (a) PSP and (b) APP. The evolution of PNSD shows clearly the
occurrence of strong nucleation and growth events on some days leading to significant increase in
CN10 and CCN0.4. During the winter months, photochemistry is relatively weak and biogenic
emissions are small. Nevertheless, our model simulations show that nucleated particles of a few
nanometers, through H₂SO₄ condensation and equilibrium uptake of HNO₃, NH₃, and H₂O, are
able to grow to 10–30 nm on most of nucleation event days and even to 60–100 nm particles that
can act as CCN during some of these days. The model captures quite well the absolute values of
CN10 ($\sim 200 - 20000$ cm$^{-3}$) as well as their daily variability at both sites, with MBE=9%, 6% and
$r =0.70$, 0.55 for the PSP and APP site, respectively. The PNSDs and CN10 time series indicate
that at both sites, CN10 is dramatically elevated (by a factor of up to $\sim 10$) in the aftermath of
nucleation events. CN10 associated with primary particles (CN10_PP, mainly black carbon and
primary organic carbon, with coating of secondary species) remains fairly constant ($\sim 100$ cm$^{-3}$)
during nucleation events. Based on the model simulation, the mean CN10 (CN10_PP) during the
two month period are 2989 (106) cm$^{-3}$ for the PSP site and 3180 (88) cm$^{-3}$ for the APP site, showing



that the secondary particles (CN10 − CN10_PP) account for >95% of total CN10. The
concentration of CCN0.4 and the fraction associated with secondary particles ($f_{CCN\_SP}$) in the
surface layer at the two sites have large variations, ranging from several tens to several thousand
per $cm^{-3}$ for CCN0.4 and ~ 0–90% for $f_{CCN\_SP}$. CCN0.4 and $f_{CCN\_SP}$ are generally elevated during
nucleation event days.

For detailed examination of the contribution of nucleation to CCN0.4 at the regional scale, a

four-day period (November 15–18, 2013, marked within a black rectangle in Fig. 3) is selected so
as to have all permutations of nucleation events and non-events at the two sites (PSP and APP).
November 15 (Day 319) has nucleation events at both sites, November 16 has nucleation event
only at PSP, November 17 has nucleation non-events at both sites, and November 18 has nucleation
event only at APP. Figure 4 shows for the NEUS, containing the PSP and APP sites, the modeled
horizontal spatial distribution of $[SO_2]$, $[H_2SO_4]$, and nucleation rate ($J$) averaged within the
boundary layer (first 7 model layers above surface). $[SO_2]$ is controlled by emission, transport,
chemistry, and deposition. Large daily variation of $[SO_2]$ in the NEUS and the important role of
$SO_2$ emission from Ohio Valley region can be clearly seen in Fig. 4. The dependence of nucleation
rate on $[H_2SO_4]$, which is determined by $SO_2$ oxidation production rate and condensation sink, is
clear over the NEUS. Consistent with the nucleation events and non-events observed at PSP and
APP sites during the 4-day period as shown in Fig. 3, Figure 4 shows that the nucleation is
generally at the regional scale with spatial distribution similar to that of $[H_2SO_4]$. These regional
wintertime nucleation events contribute significantly to CCN0.4 in the NEUS as evidenced in the
day-to-day spatial variations in CCN0.4 given in Fig. 5 (upper panels). Regions of high CCN0.4,
generally dominated by secondary particles (Fig. 5 middle panes), correspond well with areas of
high nucleation (Fig. 4, lower panels)). More than ~ 80% of CCN0.4 is of secondary origin in
regions with CCN0.4 above ~ 1000 $cm^{-3}$. Figure 5 (lower panels) also gives daily mean Cloud
Droplet Number Concentration (CDNC) in the boundary layer (liquid water content weighted
average) during the period. Apparently, clouds formed in regions of higher CCN0.4 have larger
CDNC and secondary particles contribute to CDNC in these regions, highlighting the need for
better representation of secondary particle formation and growth in regional models.

So far, our analysis focuses on aerosol and precursors near surface or in the boundary layer. To

examine the vertical variations, Figure 6 shows the two-month (November–December 2013) mean
nucleation rates and consequent contribution to CN10 (SP fraction, $f_{CN10\_SP}$) and CCN0.4 (SP



fraction, $f_{CCN\_SP}$) in the lower boundary layer (below ~ 960 mb), lower troposphere (~ 960–800
mb), middle troposphere (~ 800–470 mb), and upper troposphere (~ 470–250 mb) over the NEUS.
The model simulations indicate substantial nucleation at all altitudes although nucleation rates are
higher in lower boundary layer and upper troposphere. Horizontal distributions of nucleation rates
in lower boundary layer and lower troposphere differ significantly from those in middle and upper
troposphere, indicating quite different sources of air mass and that the influence of local emission
is limited to the lower troposphere. Secondary particles dominate CN10 at all altitudes over NEUS
and $f_{CN10\_SP}$ increases with altitudes from >~85% in lower boundary layer to >~95% in the upper
troposphere. In the lower boundary layer, secondary particles formed via nucleation contribute to
the CCN0.4 number concentration from about 20-30% over the New England region to ~ 40–50%
over the Ohio Valley region. Similar to that of CN10, the SP fraction of CCN0.4 increases with
altitudes, reaching to 50-60% in the middle troposphere and 70–85% in the upper troposphere

## 4. Summary

New particle formation (NPF) has been well recognized as an important source of ultrafine
particles which can lead to adverse health impacts and CCN which affects cloud, precipitation, and
climate. In this study, wintertime particle formation over the Northeastern United States (NEUS)
and its contribution to particle number concentrations and CCN are investigated. Wintertime NPF
in NEUS is expected to be dominated by inorganic species as a result of very low biogenic
emissions. Based on WRF-Chem-APM simulations for a two-month period (November–
December 2013) and comparisons with measurements, we show that substantial regional scale
NPF occurs in the winter over NEUS despite weaker photochemistry and low biogenic emissions.
The recently developed physics-based $H_2SO_4$-$H_2O$-$NH_3$ ternary ion-mediated nucleation scheme
appears to be able to capture the absolute values of particle number concentrations as well as their
daily variations observed at two sites in NEUS. The freshly nucleated nanometer particles can
grow to 10-30 nm on most nucleation event days and to CCN sizes during some of these days.
CN10 and CCN0.4 are dramatically elevated in the aftermath of nucleation events. Calculated
monthly mean nucleation rates in the boundary layer over the NEUS range from ~ 0.1 to ~ 2 $cm^3s^{-1}$
and spatial distributions are strongly correlated with concentration of aerosol precursors. The
monthly mean number concentrations of CN10 and CCN0.4 are around 2000–7000 $cm^{-3}$ and 100–
1000 $cm^{-3}$, respectively. Both CN10 and CCN0.4 have highest values over the Ohio Valley region,


a key source region of anthropogenic SO2. The model simulations indicate substantial nucleation
occurs at all altitudes although nucleation rates are higher in lower boundary and upper troposphere.
Secondary particles dominate CN10 at all altitudes over NEUS and its fraction increases with
altitudes from >~85% near surface to >~95% in upper troposphere. The fraction of CCN0.4 due
to secondary particles also increases with altitudes, from 20-50% in the lower boundary layer to
50-60% in the middle troposphere and 70–85% in the upper troposphere.
**Data availability.** The model output and observational data used for comparison are available on
request from the authors.
**Author contributions.** FY, GL, and YZ developed the project idea. GL and FY updated the model
and carried out the numerical simulations. FY and AN wrote the paper, with contribution from GL
and JJS. JJS and JBS contributed observational data used in the comparison.
**Competing interests.** The authors declare that they have no conflict of interest.
**Acknowledgements**: This study was supported by NYSERDA under contract 100416 and NSF
under grants OISE-1545917 and AGS-1550816, and. Ammonia Monitoring Network (AMoN)
data used for comparison is from National Atmospheric Deposition Program (NRSP-3), 2017,
NADP Program Office, Illinois State Water Survey, University of Illinois, Champaign, IL 61820
(http://nadp.sws.uiuc.edu/AMoN/).

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

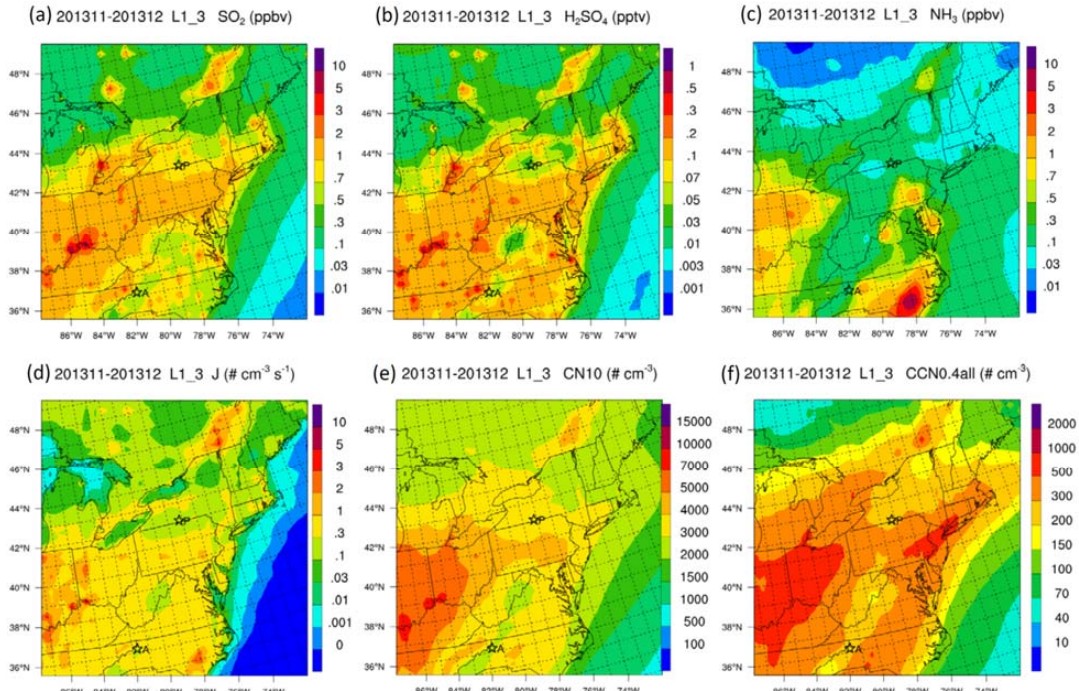


**Figure 1.** Horizontal spatial distribution of WRF-Chem-APM simulated average wintertime (2013
November–December) **(a)** [SO$_2$], **(b)** [H$_2$SO$_4$], **(c)** [NH$_3$], **(d)** nucleation rate (*J*), **(e)** number
concentration of condensation nuclei > 10 nm (CN10), and **(f)** cloud condensation nuclei at
supersaturation 0.4% (CCN0.4) in the lower boundary layer (~ 0 – 400 m above surface, first three
model layers) over the Northeastern United States (NEUS). Measurement sites Appalachian State
University (APP), North Carolina (A) and Pinnacle State Park (P), New York are marked on the
maps.

397

398

**Figure 2.** Modeled diurnal variability of wintertime (November–December 2013) (a) temperature (T) and relative humidity (RH), (b) wind direction (WD) and solar radiation (SR), (c) [SO$_2$] and precipitation, (d) [NH$_3$], [H$_2$SO$_4$], and concentration of low-volatile secondary organic gas ([LV-SOG]), and (e) nucleation rate (J) and CN10 at the Pinnacle State Park (PSP) site compared with in situ measurements. X-axis is the day of year (DOY).

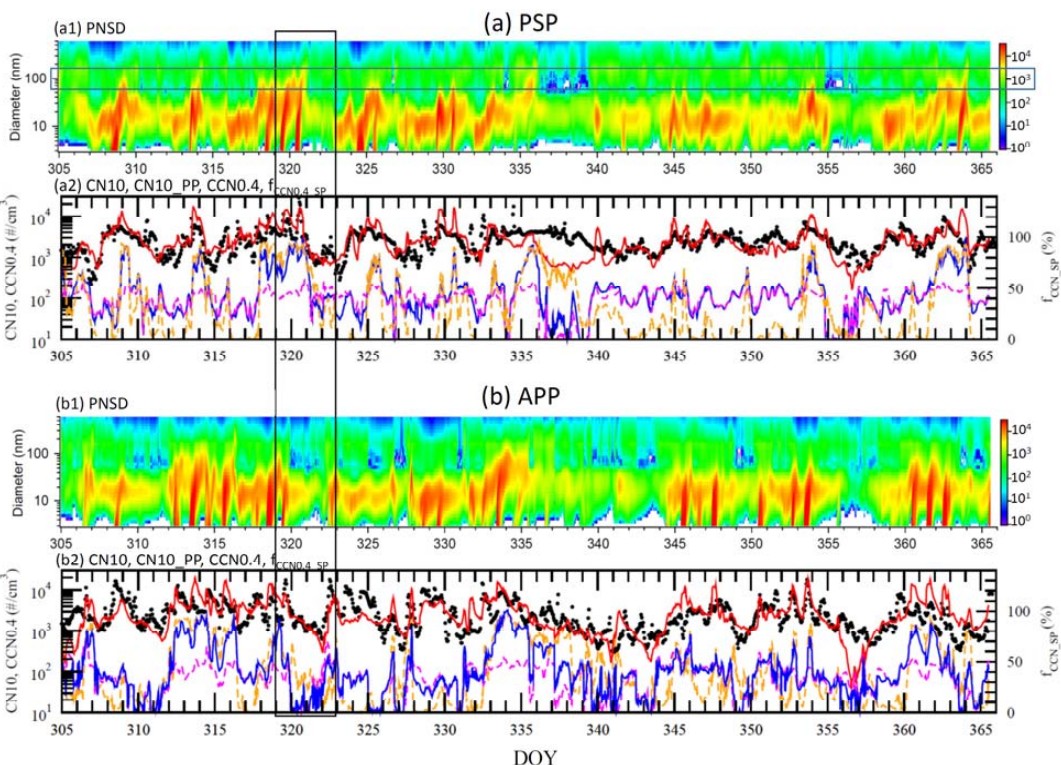

**Figure 3.** For the (a) PSP and (b) APP sites in the NEUS: Modeled wintertime (November–December 2013) evolution of particle number size distributions (PNSD, a1, b1), and time series (a2, b2) of CN10 (red line), CN10 due to primary particles (CN10_PP, dashed magenta line), CCN0.4 (blue line), and percentage of CCN0.4 associated with secondary particles ($f_{CCN\_SP}$, dashed orange line). In a2 and b2, CN10 values from observations (black circles) are also shown for comparison. The model results are for the model surface layer (~0–100 m above surface). Selected 4-day period from November 15–18, 2013 with nucleation events and non-events is marked within a black rectangle.





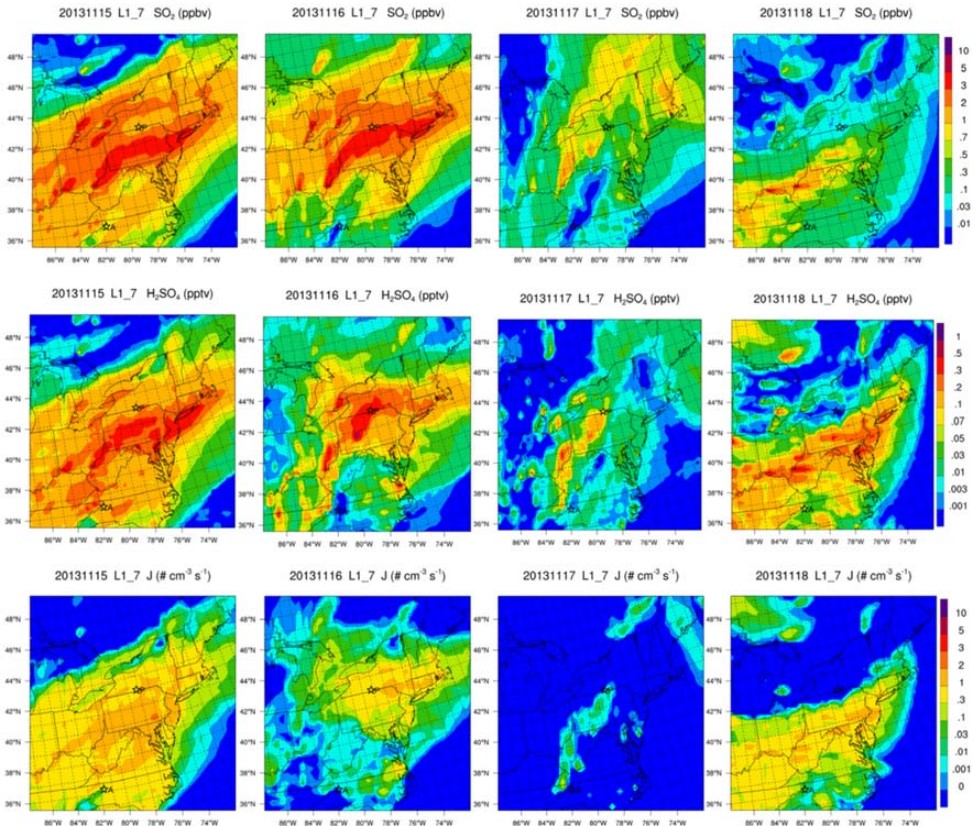

**Figure 4.** For the each of the 4-day period from (left to right) November 15–18, 2013: (top to bottom) modeled horizontal spatial distribution of [SO₂], [H₂SO₄], and nucleation rate ($J$) over the NEUS, with the measurement sites Pinnacle State Park (P) and APP (A) marked on the maps.

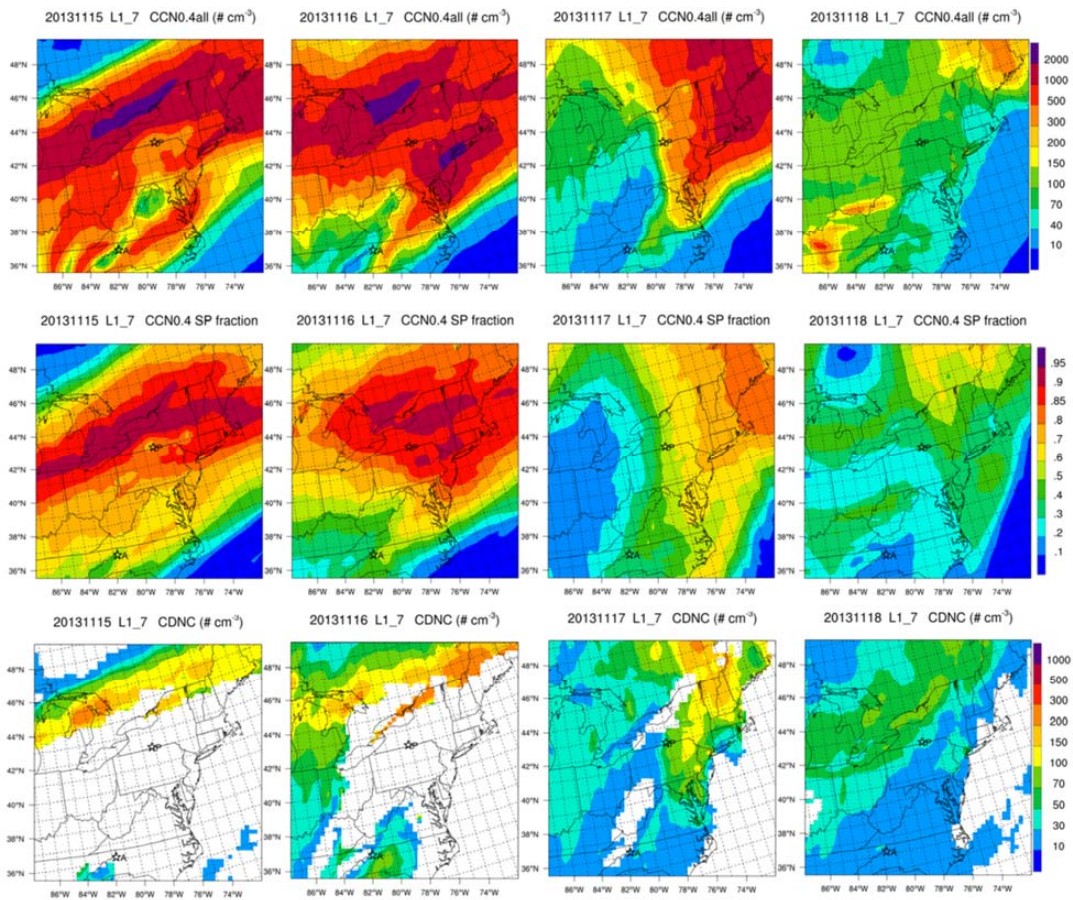

**Figure 5.** For the each of the 4-day period from (left to right) November 15–18, 2013: (top) CCN0.4 and (middle) its secondary particle fraction (CCN0.4 SP), and (bottom) cloud droplet (CDNC) modeled horizontal spatial distribution over the NEUS, with the measurement sites Pinnacle State Park (P) and APP (A) marked on the maps.

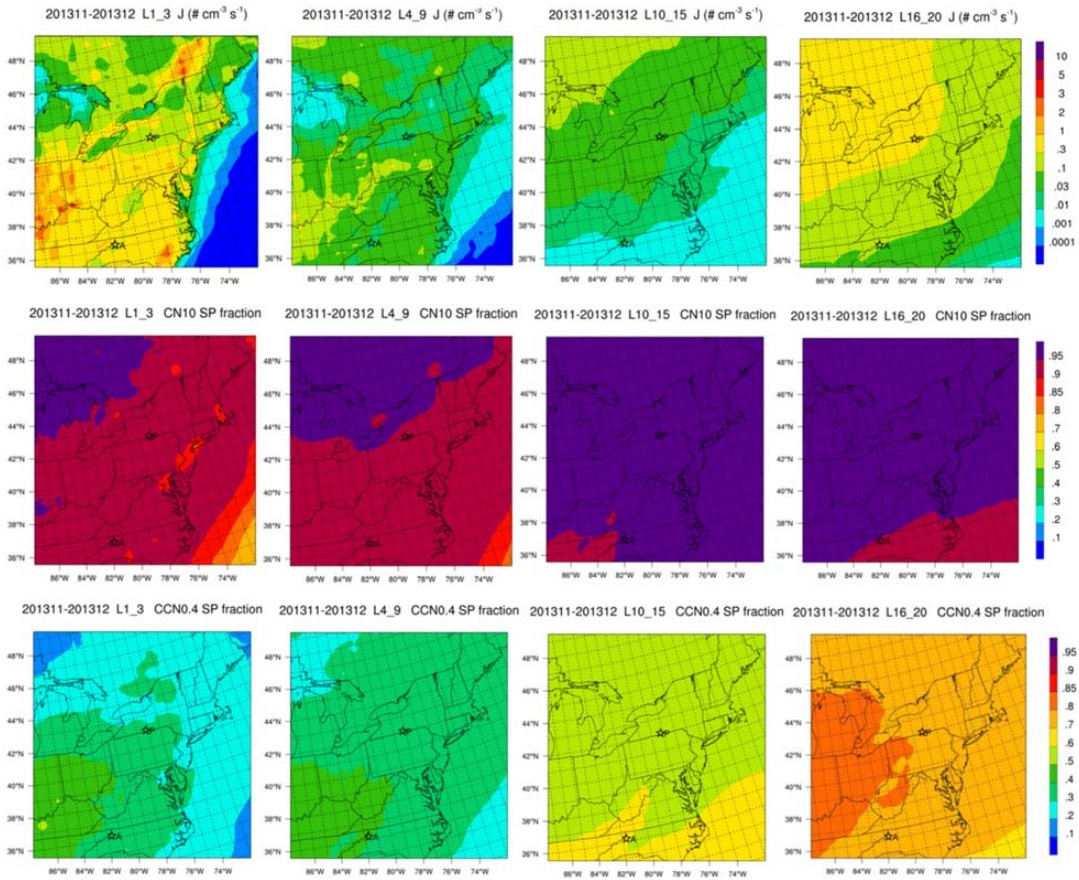

**Figure 6.** Modeled average wintertime (2013 November–December) (top) nucleation rate (*J*), (bottom) CN SP fraction, and (bottom) CCN0.4 SP fraction for (left to right) the surface layer, lower, middle, and upper troposphere.