# Peer review of "Wintertime New Particle Formation and Its Contribution to Cloud Condensation"

_Atmospheric Chemistry and Physics, 2019_

## Referee Comment (RC1) · Anonymous Referee #1 · 6 Nov 2019

This paper describes winter time NPF in the northeastern United states using the modeling predictions combined with ambient measurements of aerosol size distributions made at two sites and their contribution to CCN production. NPF is usually considered as a main source of CCN. And observations have shown that NPF usually takes place less frequently during the winter, in many locations over the world. So the results presented in this paper are interesting considering these factors. This is a well-written paper, easy to read.

Perhaps, the model used too high sulfuric acid concentrations to predict NPF winter? We made long-term measurements of NPF and sulfuric acid in Ohio and our measure-

ments show that winter time sulfuric acid does not exceed 3e6 per cubic centimeter (Erupe et al., 2010; Yu et al., 2014). And this paper claims that in order to have NPF, sulfuric acid higher than 3e6 per cubic centimeter is needed.

Related to this, at the same site in Ohio, we also found lower frequency of NPF during the winter than other seasons, very likely due to low sulfuric acid concentrations (Erupe et al., 2010; Kanawade et al., 2012).

Does the model consider temperature effects on nucleation and growth? NPF becomes more favorable at lower temperatures, as shown from laboratory studies (Duplissy et al., ; Yu et al., 2017; Tiszenkel et al., 2019). If the model includes this feature, then maybe this is due to lower temperatures?

It would be nice to give some explanation why ternary ion nucleation (as opposed to neutral ternary nucleation) is important? What are the potential sources of ions in winter in the boundary layer?

And is it possible to explain the growth rate from 3 nm to the CCN size with the sulfuric acid and ammonia? If not, what makes new particles grow so fast to become CCN?

References:

Duplissy, J., et al. (2016), Effect of ions on sulfuric acid-water binary particle formation: 2. Experimental data and comparison with QC-normalized classical nucleation theory, J. Geophys. Res., 121, 1752-1775.

Erupe, M. E., et al. (2010), Correlation of aerosol nucleation rate with sulfuric acid and ammonia in Kent Ohio: an atmospheric observation, J. Geophys. Res., 115, Doi:10.1029/2010JD013942.

Kanawade, V., D. R. Benson, and S. H. Lee (2012), Statistical analysis of 4 year measurements of aerosol sizes in a semi-rural U.S. continental environment, Atmos. Environ., 59, 30-38.

[Figure]

Tiszenkel, L., et al. (2019), Temperature effects on sulfuric acid aerosol nucleation and growth: initial results from the TANGENT study, Atmos. Chem. Phys., 19, 8915-8929.

---

## Referee Comment (RC2) · Anonymous Referee #2 · 7 Nov 2019

This manuscript investigates the contribution of nucleation to particle number and CCN in the eastern US using WRF-Chem-APM. The simulations show the majority of the BL number and around half of CCN0.4% from nucleation. The simulated CN10 and some gases were evaluated against measurements at 2 sites. I'm in favor of publication once some issues have been addressed.

Abstract and other places: There are statements about how the nucleation is entirely inorganic because of low biogenic emissions in winter. However, while this is a sound hypothesis, it was not explicitly tested. Please weaken the language to make it clear that the lack of organic nucleation was assumed, not a finding.

[Figure]

Abstract and throughout (e.g. L60-61): Please add more statements of "The model shows..." or "We predict..." etc. The current writing style likely has these statements implied, but there is a risk of this sentiment being missed by some readers, and they may think this was more than a model finding..

L25-27: This sentence is strange. What is the changing paradigm of wintertime precip? This isn't discussed in the paper other than maybe one sentence at the end of the intro (L61-65, though it doesn't refer to a changing paradigm).

L54-56: The statement seems incomplete. I believe the conclusion of Yu et al. (2015) was that the ion-mediated scheme they used did not have a temperature dependence, which caused it to overpredict in the summer. Yu et al. (2017) estimates a correction for the temperature dependence that may prevent the overprediction in the summer. The current statement should explain the findings better.

Not that Fangqun won't know the reference, but for completeness: Yu, F., Luo, G., Nadykto, A. B., and Herb, J.: Impact of temperature dependence on the possible contribution of organics to new particle formation in the atmosphere, Atmos. Chem. Phys., 17, 4997-5005, https://doi.org/10.5194/acp-17-4997-2017, 2017

L132-134: Please add the specific instruments from which data was used here.

Figure 2d: It would be useful to show the NH3 values from the model averaged over the times of the AMoN site.

L177: [NH3] *partitioning* is calculated with ISOROPIA II

L215: The abstract said >85% for the surface

Figure 3: It's confusing that there is a line for CN10 due to primary particles and CCN0.4 due to secondary particles. Please make them either both primary or both secondary for consistency.

L240: "Apparently" doesn't seem like the right word here. It makes this seem like the

CCN-CDNC connection was not expected.

L242: Why does it highlight the need for *better* representation. Has this paper found deficiencies in representation? I don't think this paper has evaluated this.

---

## Author Comment (AC1) · 28 Jan 2020

The authors would like to thank the reviewer for the constructive comments. Our replies to the comments are given below, with the original comments in black, and our response in blue. We have revised the manuscript accordingly. All changes made to the manuscript have been marked with Track-Change tool in one of submitted files.

**Anonymous Referee #1**
This paper describes winter time NPF in the northeastern United states using the modeling predictions combined with ambient measurements of aerosol size distributions made at two sites and their contribution to CCN production. NPF is usually considered as a main source of CCN. And observations have shown that NPF usually takes place less frequently during the winter, in many locations over the world. So the results presented in this paper are interesting considering these factors. This is a well-written paper, easy to read.
We appreciate the referee's positive comments about this work.

Perhaps, the model used too high sulfuric acid concentrations to predict NPF winter?
We made long-term measurements of NPF and sulfuric acid in Ohio and our measurements show that winter time sulfuric acid does not exceed 3e6 per cubic centimeter (Erupe et al., 2010; Yu et al., 2014). And this paper claims that in order to have NPF, sulfuric acid higher than 3e6 per cubic centimeter is needed.
Related to this, at the same site in Ohio, we also found lower frequency of NPF during the winter than other seasons, very likely due to low sulfuric acid concentrations (Erupe et al., 2010; Kanawade et al., 2012).
This is a very good point. We agree that sulfuric acid concentrations ($[H_2SO_4]$) are critical for NPF. The model calculates $[H_2SO_4]$ through the balance between the photochemical production ($SO_2+OH$) and condensation. The model appears to generally capture the observed values and variations of $SO_2$ and solar radiations for the PSP site (Fig. 2b and Fig. 2c), both are important for the photochemical production of $H_2SO_4$. Based on the model prediction, $[H_2SO_4]$ can reach ~ $1E7/cm^3$ or higher and it is during these days that nucleation is significant (Figs. 2d and 2e). As emphasized in the paper, these nucleation events are necessary to explain the observed increase in CN10 (Fig. 2e).

It is true that the model predicted $[H_2SO_4]$ is higher than those observed (with CIMS) in Kent, Ohio during the winter (Erupe et al., 2010; Yu et al., 2014). The possible reasons for the difference remain to be investigated. One possible explanation is the well-recognized 1-2 orders-of-magnitude lower concentrations of sulfuric acid monomer measured with CIMS than the total-sulfate values measured with MARGA and the theoretical values calculated from the vapor pressure of sulfuric acid (Neitola et al., 2015).

We have added a discussion on this in the revised text:

"It should be noted that the model predicted $[H_2SO_4]$ is higher than those observed with a chemical ionization mass spectrometer (CIMS) during the winter in Kent, Ohio (Erupe et al., 2010; Yu et al., 2014), also located in the NEUS where wintertime nucleation was observed to occur on ~17% of days (Kanawade et al., 2012). The possible reasons for the difference of model-predicted and CIMS-observed $[H_2SO_4]$ remain to be investigated. One possible explanation is that sulfuric acid molecules are bonded with base molecules (e.g. ammonia and

amines), leading to the well-recognized 1–2 orders-of-magnitude lower concentrations of sulfuric acid monomers measured with CIMS than the total-sulfate values measured with MARGA and the theoretical values calculated from the vapor pressure of sulfuric acid (Neitola et al., 2015)."

Does the model consider temperature effects on nucleation and growth? NPF becomes more favorable at lower temperatures, as shown from laboratory studies (Duplissy et al., ; Yu et al., 2017; Tiszenkel et al., 2019). If the model includes this feature, then maybe this is due to lower temperatures?

Yes. Temperature is one of the most important parameters controlling NPF. Nucleation is favored at lower temperatures but other factors ($[H_2SO_4]$, $[NH_3]$, ionization rates, etc.) are also important. We have added a sentence emphasizing this point.

It would be nice to give some explanation why ternary ion nucleation (as opposed to neutral ternary nucleation) is important? What are the potential sources of ions in winter in the boundary layer?

The main reason is that charged clusters have a lower nucleation barrier and thus ternary ion nucleation is favored as opposed to neutral ternary nucleation. The details can be found in the reference (Yu et al., 2018) cited in the paper. The main sources of ions in winter in the boundary layer include galactic cosmic rays and radioactive materials from soils. We have added two sentences in the first paragraph of the Results session.

And is it possible to explain the growth rate from 3 nm to the CCN size with the sulfuric acid and ammonia? If not, what makes new particles grow so fast to become CCN?

As pointed out in the text (line 226 in the changes tracked version), equilibrium uptake of $HNO_3$ also contributes to particle growth in the winter.

---

## Author Comment (AC2) · 28 Jan 2020

The authors would like to thank the reviewer for the useful comments which help to improve the manuscript. Our replies to the comments are given below, with the original comments in black, and our response in blue. We have revised the manuscript accordingly. All changes made to the manuscript have been marked with Track-Change tool in one of submitted files.

**Anonymous Referee #2**
This manuscript investigates the contribution of nucleation to particle number and CCN in the eastern US using WRF-Chem-APM. The simulations show the majority of the BL number and around half of CCN0.4% from nucleation. The simulated CN10 and some gases were evaluated against measurements at 2 sites. I'm in favor of publication once some issues have been addressed.

Abstract and other places: There are statements about how the nucleation is entirely inorganic because of low biogenic emissions in winter. However, while this is a sound hypothesis, it was not explicitly tested. Please weaken the language to make it clear that the lack of organic nucleation was assumed, not a finding.

Abstract and throughout (e.g. L60-61): Please add more statements of "The model shows. . ." or "We predict. . ." etc. The current writing style likely has these statements implied, but there is a risk of this sentiment being missed by some readers, and they may think this was more than a model finding..

We would like to clarify that the lack of organic nucleation was based on model simulations, not assumptions. The model calculates biogenic emissions based on MEGAN (see Section 2.1). Yes, we have revised the relevant sentences as suggested to weaken the language.

L25-27: This sentence is strange. What is the changing paradigm of wintertime precip? This isn't discussed in the paper other than maybe one sentence at the end of the intro (L61-65, though it doesn't refer to a changing paradigm).

This is a valid point. We have deleted this sentence from the abstract.

L54-56: The statement seems incomplete. I believe the conclusion of Yu et al. (2015) was that the ion-mediated scheme they used did not have a temperature dependence, which caused it to overpredict in the summer. Yu et al. (2017) estimates a correction for the temperature dependence that may prevent the overprediction in the summer. The current statement should explain the findings better.

Not that Fangqun won't know the reference, but for completeness: Yu, F., Luo, G., Nadykto, A. B., and Herb, J.: Impact of temperature dependence on the possible contribution of organics to new particle formation in the atmosphere, Atmos. Chem. Phys., 17, 4997-5005, https://doi.org/10.5194/acp-17-4997-2017, 2017

Yes, we have revised the statements to include the results of the 2017 paper.

L132-134: Please add the specific instruments from which data was used here.

Added as suggested.

Figure 2d: It would be useful to show the NH3 values from the model averaged over the times of the AMoN site.

Added as suggested.

L177: [NH3] *partitioning* is calculated with ISOROPIA II
Modified as suggested.

L215: The abstract said >85% for the surface
The value given here is for the two specific sites (PSP and APP) while >85% given in the abstract is for the whole NEUS region.

Figure 3: It's confusing that there is a line for CN10 due to primary particles and CCN0.4 due to secondary particles. Please make them either both primary or both secondary for consistency.
We have changed CCN0.4 due to SP to CCN0.4 due to PP.

L240: "Apparently" doesn't seem like the right word here. It makes this seem like the CCN-CDNC connection was not expected.
This word has been deleted from the sentence.

L242: Why does it highlight the need for *better* representation. Has this paper found deficiencies in representation? I don't think this paper has evaluated this.
We have changed "better" to "proper".